# Evaluating the Safety and Efficacy of Transcranial Pulse Stimulation on Autism Spectrum Disorder: A Double-Blinded, Randomized, Sham-Controlled Trial Protocol

**DOI:** 10.3390/ijerph192315614

**Published:** 2022-11-24

**Authors:** Teris Cheung, Yuen Shan Ho, Kwan Hin Fong, Yuen Ting Joyce Lam, Man Ho Li, Andy Choi-Yeung Tse, Cheng-Ta Li, Calvin Pak-Wing Cheng, Roland Beisteiner

**Affiliations:** 1School of Nursing, The Hong Kong Polytechnic University, Hong Kong SAR 999077, China; 2The Mental Health Research Centre, The Hong Kong Polytechnic University, Hong Kong SAR 999077, China; 3Department of Psychiatry, The Chinese University of Hong Kong, Hong Kong SAR 999077, China; 4Department of Health and Physical Education, The Education University of Hong Kong, Hong Kong SAR 999077, China; 5Department of Psychiatry, School of Medicine, National Yang Ming Chiao Tung University, Taipei 112, Taiwan; 6Department of Psychiatry, The University of Hong Kong, Hong Kong SAR 999077, China; 7Department of Neurology, Medical University of Vienna, 1090 Wien, Austria

**Keywords:** efficacy, transcranial pulse stimulation, RCT, autism spectrum disorder, neuromodulation, adolescents

## Abstract

Autistic spectrum disorder (ASD) is a common developmental disorder in children. The latest non-intrusive brain stimulation (NIBS) technology—transcranial pulse stimulation (TPS)—has been proven effective in older adults with mild neurocognitive disorders and adults with major depressive disorder. Nonetheless, there is so far no robust randomized controlled trial (RCT) conducted on adolescents with ASD nationwide. This study proposes a two-armed (verum TPS group vs. sham TPS group), double-blinded, randomized, sham-controlled trial. Both groups will be measured at four timepoints, namely, baseline (T1), 2 weeks immediately after post-TPS intervention (T2), and at the 1-month (T3) and 3-month (T4) follow-ups. Thirty-four subjects, aged between 12 and 17, diagnosed with ASD will be recruited in this study. All subjects will be computerized randomised into the verum TPS group or the sham TPS group on a 1:1 ratio. All subjects will undertake functional MRI (fMRI) before and after the 2-weeks TPS interventions, which will be completed in 2 weeks’ time. This will be the first RCT evaluating the efficacy of TPS adolescents with ASD in Hong Kong. Clinical Trial Registration: ClinicalTrials.gov, identifier: NCT05408793.

## 1. Introduction

Autistic spectrum disorder (ASD) is a neurodevelopmental disorder characterized by impairment in reciprocal social interaction, language dysfunction, and restricted interests, associated with behaviour problems. The prevalence is 62 in 10,000 in the general population [1]. ASD is often linked with functional impairment, and increased risk of psychiatric and medical morbidity [2]. Furthermore, ASD causes significant distress and mental health-related problems in the caregivers of people with ASD [3]. These negative detrimental effects will inevitably increase the global disease burden on our health care system, and thus, there is a pressing need to formulate evidenced-based, robust interventional studies to restore optimal wellbeing for this at-risk, vulnerable population.

Meanwhile, the exact underlying pathophysiology of ASD remains unknown. There was no effective and specific pharmacological treatment for core symptoms of ASD [4]. Conventional treatment for ASD consists of behavioural intervention and social skill training; these treatment options are time-consuming and labour-intensive. Therefore, a new, innovative but effective treatment alternative that is well-tolerated by ASD adolescents is warranted.

Transcranial pulse stimulation (TPS) is the latest NIBS used to stimulate the brain by inducing ultrashort ultrasound waves to the treatment region of the brain. TPS has been proven effective in the treatment of Alzheimer’s disease in a recent multicentre European study [5]. Study findings reported significant improvements in memory and cognition, a reduction in depressive symptoms [6], and a reduction in cortical atrophy [7]. Another local pilot open-label RCT also proved that TPS is effective in significantly reducing depression severity in adults diagnosed with major depressive disorder [8] in Hong Kong. Nonetheless, there is so far no further study using TPS evaluating the efficacy and safety on young adolescents with ASD. This research gap has given us the impetus to execute this trial.

### 1.1. Objectives

The aims of this study will be (1) to evaluate the efficacy and tolerability of TPS on adolescents (age 12–17) with ASD in Hong Kong; (2) to examine the association between TPS and ASD core symptom severity, executive functions, and social functions; and (3) to examine the brain functional connectivity changes immediately after the 2-week TPS treatment via neuroimaging.

### 1.2. Hypotheses

#### 1.2.1. Primary Hypothesis

Participants in the verum TPS group will have a significant reduction in Childhood Autism Rating Scale (CARS) score at posttreatment compared with the sham TPS group and be maintained at the 1-month and 3-month follow-ups.

#### 1.2.2. Secondary Hypotheses

Participants in the verum TPS group will have significant improvement in the Autism Spectrum Quotient (AQ) compared with the sham TPS group, which will be maintained at the 1-month and 3-month follow-ups.Participants in the verum TPS group will have significant improvements in the Social Responsiveness Scale (SRS) compared with the sham TPS group, which will be maintained at the 1-month and 3-month follow-ups.Participants in the verum TPS group will have significant improvement in the Faux Pas Recognition Test compared with the sham TPS group, which will be maintained at the 1-month and 3-month follow-ups.Participants in the verum TPS group will have significant improvement in the Stroop test compared with the sham TPS group, which will be maintained at the 1-month and 3-month follow-ups.Participants in the verum TPS group will have significant improvement in the Clinical global impression—severity and improvement scale (CGI-S and CGI-I) compared with the sham TPS group, which will be maintained at the 1-month and 3-month follow-ups.Participants in the verum TPS group will have significant brain function connectivity changes at posttreatment fMRI and resting-state MRI compared to the sham TPS group, which will be maintained at the 1-month and 3-month follow-ups.

## 2. Materials and Methods

### 2.1. Trial Design

This proposed study is a two-armed, randomized, double-blind, sham-controlled trial evaluating the efficacy and tolerability of a 2-week TPS treatment on ASD among adolescents. This study’s trial design strictly complies with the Consolidated Standards of Reporting Trials (CONSORT) statement [9] and Good Clinical Practice. Participants will be randomly allocated into the verum TPS group or sham TPS group. All the participants’ parents will be informed about the randomization procedures and that they have a 50% chance of receiving the verum TPS or the sham TPS. This study will be conducted in accordance with the Declaration of Helsinki [10]. Both groups will be measured at baseline (T1), immediately after the 2-week intervention (T2), and at 1-month (T3) and 3-month (T4) follow-ups [11] (Figure 1). We registered this trial with ClinicalTrials.gov on 7 June 2022 (NCT05408793).

### 2.2. Subjects

Participants will be recruited using a QR code flyer embedded with an online application form. This QR code flyer will also be flagged up in communal areas on campus in the research sites and collaborative universities including the Hong Kong Polytechnic University (HKPU), University of Hong Kong (HKU), and The Educational University of Hong Kong (EdUHK). The recruitment period will span three months. Participants will require parental written consent for TPS treatment and neuroimaging. Both participants and their parents will be informed that the TPS treatment study will involve an off-label treatment, and the possible risks and side effects will be clearly explained in the Information Sheet.

### 2.3. Inclusion Criteria

To be eligible, participants should (1) be 12–17 years of age, (2) be of Chinese ethnicity, (3) have an ASD diagnosis according to the 5th Edition of the Diagnostic and Statistical Manual of Mental Disorders (DSM-5), (4) not have any change in their drug regime, and (5) currently take prescribed psychotropic medications for ≥3 months.

### 2.4. Exclusion Criteria

Participants who (1) have a DSM-5 diagnosis other than ASD; (2) have concomitant major medical conditions; (3) have neurological problems (e.g., brain tumour or brain aneurysm); (4) have haemophilia or other clotting disorders, or thrombosis; (5) have a metal implant in the brain/brain treated region; (6) undertake corticosteroid treatment in the past 6 weeks prior to the first TPS treatment; or (7) have a Childhood Autism Rating Scale (CARS) score of 30 or below (i.e., no ASD) will be excluded from this study.

### 2.5. Sample Size

Based on a similar RCT design (4), the estimated effect size of repeated transcranial magnetic stimulation is 0.5. Nonetheless, this study is the first study evaluating TPS on ASD. Hence, we hypothesize a small effect size (0.3) on this pilot study. We used G*Power 3.1.9.4 to estimate the sample size for this study. Considering a similar study using rTMS with an effect size of 0.5, we decided to use 0.5 effect size in this first study. Thus, the sample size required is 34 in this study (17 subjects in each group).

### 2.6. Screening and Self-Administered Questionnaire

Participants’ parents will complete a QR code online application form soliciting sociodemographic information (age, gender, educational background, monthly family household income, living circumstances, school year, participant’s psychiatric history and duration of ASD diagnosis (in years/months), age of diagnosis, duration of taking prescribed medications (in years/months), current drugs and dosages, and family history of psychiatric disorder).

Eligible subjects will then fill in the online screening tool (Childhood Autism Rating Scale) (CARS); those with a CARS total score > 30 will be recruited. Subjects’ medical history, treatment regime, and developmental history will be obtained by direct enquiry with subjects’ parents by zoom interview prior to neuroimaging and TPS treatment. Both participants and parents will be interviewed by the principal investigator (PI) and the research personnel. Parents need to hold a valid medical certificate of his/her children’s ASD diagnosis (year/month) and show the most recent prescribed formulation sheet to the research team during the online interview. Any parent who fails to show these documentary proofs will not be invited to participate in this trial.

### 2.7. Randomization, Allocation, and Masking

Block randomisation will be used in this study. Each block will consist of 6 participants at a 1:1 ratio between the two groups (total: 6 blocks). In the randomisation procedure, each participant will be assigned a serial reference number generated by a computer. These numbers will not be decoded until the intervention group is assigned. An offsite statistician who is an independent team member, who will not be involved in the enrolment, intervention, or assessments, will perform the randomization process. Participants and research assistants who will be responsible for assessment and data analysis will be blinded to the group allocation (Figure 1). Participants will be asked to guess their treatment allocation in the 6th TPS session to determine the success of the blinding [12].

### 2.8. Interventions

The TPS intervention venue will be located at the Integrative Health Clinic (IHC), the Hong Kong Polytechnic University. A licensed mental health practitioner (PI: Dr. Teris Cheung) will deliver the TPS intervention.

#### 2.8.1. Intervention TPS

The TPS system consists of a mobile single transducer and an infrared camera system, which incorporates neuro-navigation; this system was developed by NEUROLITH, Storz Medical AG, Tägerwilen, Switzerland. This TPS system can generate single ultrashort (3 µs) ultrasound pulses with 0.2–0.25 energy levels (mJ/mm^2^) and 2.5–4 Hz pulse frequencies (pulses per second). During the TPS session, participants will sit in an adjustable electronic chair and wear a BodyTrack system that consists of a tracking glass with markers, a 3D camera, and a TPS handpiece. The reasons for wearing this BodyTrack system are to ensure that participant’s heads match their own fMRI (T1 images) so that the interventionist can visualize each pulse applied and documented in real time. The real-time tracking of the handpiece position enables automatic visualization of the treated brain region, highlighted in green.

#### 2.8.2. Treatment Brain Region

We will target the right temporoparietal junction (rTPJ), a key node for social cognition, which is a typical area of difficulty among individuals with ASD [13]. We selected the treatment brain region based on previous research [4], which had demonstrated abnormal brain activation in ASD and can improve social communication in adolescents and young adults with ASD. In this study, we will deliver 800 pulses (confirmed with the NEUROLITH-TPS manufacturer, neurologist, and psychiatrist in the project team) in each TPS session to subject’s rTPJ. Each TPS session will take approximately 30 min to complete. The interventionist will manually move the TPS handpiece over the subject’s skull to visualize the TPS pulses in their T1 brain images. The treatment will be carried out by trained staff holding the applicator in their hand. All TPS treatment sessions will be recorded for individual evaluation of the intracerebral pulse localizations.

#### 2.8.3. Sham–Control TPS

The procedure for sham stimulation will be identical, except that the castor oil used in the TPS group will be replaced by air-filled stand-off cushion in the hand piece. This sham device will be able to produce similar sound and stimuli on the subject’s head. Participants will be asked to continue their routine medication regimen throughout TPS intervention period.

#### 2.8.4. Intervention Dose

All eligible participants will be randomized to receive 6 verum TPS treatment sessions or sham TPS, with 3 sessions per week on alternate days, and for two consecutive weeks. Each TPS session will last for 20 min. The outcome measurements will be assessed at baseline, immediately after intervention (2 weeks), and at 1-month and 3-month F/Us (Figure 1, CONSORT flow diagram).

### 2.9. Fidelity

The project team will ascertain whether the interventions are delivered as intended in the protocol. The interventionist (PI) obtained a PhD in Social Sciences (HKU) and is a UK and HK licensed mental health professional with more than 10 years’ clinical experience in mental health and neuroscience. The research associate will issue WhatsApp message reminders (e.g., TPS intervention schedule, fMRI scan appointments, and f/u appointments slips) to parents and monitor subjects’ progress, adverse effect, and adherence throughout the trial period.

### 2.10. Adverse Effects and Risk Indicators

A TPS adverse effect associated with TPS administration will be generated from existing literature to monitor tolerability and adverse events throughout the TPS intervention period. NIBS is also considered very safe for paediatric populations, as evidenced by a recent study findings that showed no adverse effects across 382 young children under 18 years [14].

### 2.11. Ethical and Data Security Considerations

Ethical approval will be sought from the Human Subjects Ethics Sub-committee, the Hong Kong Polytechnic University. This study adheres the ethical principle guidelines written by the Declaration of Helsinki developed by the World Medical Association. All eligible participants in this study will be covered by trial insurance. Potential risks associated with fMRI and TPS will be clearly indicated in the Information Sheet and written consent form. Right of withdrawal, voluntary participation, participant anonymity, and confidentiality will be respected. Since participants are <18 years old, their parents will have to provide written consent in this study.

### 2.12. Outcome Evaluation (Primary and Secondary Outcomes)

#### 2.12.1. Baseline Assessment

##### Demographic Data

The subjects’ basic demographic data, including age, gender, body mass index, years of education, birth history, number of siblings, financial condition of family, and family history of ASD, will be obtained. Subjects’ psychiatric history, age of diagnosis, medical health history, current treatment regime and compliance, and developmental history will be solicited by direct face-to-face interviews with the participants’ parents.

#### 2.12.2. Primary Outcome

The primary outcome will be assessed by the Childhood Autism Rating Scale (CARS). The CARS is a 15-item behavioural rating scale developed to identify autism and to examine its severity. The 15 items include different domains (e.g., relating to people, imitative behaviour, emotional response, body and object use, adaptation to change, visual/listening/perceptive response, fear or anxiety, verbal/nonverbal communication, activity level, consistency of intellective relations, and general impressions [15]. Total scores range from 15 to 60; scores below 30 indicate non-autistic range, scores between 30 and 36.5 indicate mild to moderate autism, and scores from 37 to 60 indicate severe autism [16]. The assessment tool is well-validated and has been widely used in various studies. CARS will be assessed at baseline, immediately after post-stimulation at week 2, and at 1-month, and 3 months post-stimulation follow-up.

#### 2.12.3. Secondary Outcomes

##### Autism Spectrum Quotient (AQ)—Adolescent Version

It is a self-report instrument for autistic traits. The AQ score ranges from 0 to 50. AQ is made up of 10 questions assessing (1) social skill, (2) attention switching, (3) attention to detail, (4) communication, and (5) imagination [17].

##### The Social Responsiveness Scale (SRS)

SRS is an instrument measuring the continuum of autism symptom severity, which is commonly used in children and adolescents between the ages of 4 and 18 years [18]. SRS consists of 65 items subsumed in five “a priori” content areas of social deficits, i.e., social awareness, social cognition, social communication, social motivation, and autistic mannerisms. Parents rate each item on a 4-item Likert scale (0–4). The higher the scores, the more severe the social deficits. Multiple studies have demonstrated that the SRS has satisfactory reliability and validity for measuring autism symptoms in individuals < 18 years of age [19,20].

##### Australian Scale for Asperger’s Syndrome (ASAS)

ASAS is used to screen the behaviours and abilities indicative of Asperger’s syndrome in subjects older than 6 years old [21]. It has 25 items, and parents/teachers/professionals who know the child can rate these aspects using the Chinese version.

##### Faux Pas Recognition Test (FPRT)

It is a very common advanced test to measure the theory of mind and social cognition [22].

##### Stroop Test

It is a neuropsychological test commonly used to assess the inhibition control component of executive function. It tests the subject’s ability to inhibit cognitive interference that occurs when the processing of a specific stimulus feature impedes the simultaneous processing of a second stimulus attribute [23].

##### Working Memory

The working memory of participants will be measured with the memory span tasks (both forward and backward digit recall). The digit span test has been used previously to assess working memory in school-aged children with ASD in Hong Kong.

##### Clinical Global Impression—Severity and Improvement Scale (CGI-S and CGI-I)

CGI-S is a 7-point clinician rating scale based upon observed and reported symptoms, behaviour, and function in the past seven days. Clearly, symptoms and behaviour can fluctuate over a week; the score should reflect the average severity level across the seven days. CGI-I requires the clinician to assess the extent to which the subject’s illness improved or worsened, when compared to baseline before the intervention. The two rating scales are usually used as a combination to supplement each other [24].

##### Neuroimaging

Participants will have to undergo neuroimaging twice (pre-and posttreatment MRI) to evaluate any changes in structural and functional brain connectivity changes. Structural MRI, DTI, and rs-fMRI were taken using a 3T Tesla scanner at The Hong Kong Polytechnic University. The whole scan will take about 45 min. Structural MRI scans (T1) will be used to assess regional volume differences across the whole brain. High-resolution sagittal 3D T1-weighted (MPRAGE) images of 1 × 1 × 1mm will be acquired with repetition time (TR) = 1820 ms, echo time (TE) = 3.75 ms, inversion time (TI) = 1100 ms, and flip angle = 70°. DTI sequencing will be conducted using single-shot spin-echo echo-planar imaging, with diffusion-sensitizing gradients applied along 16 non-collinear directions with diffusion weighting factor b = 1000 s/mm^2^, plus two b = 0 images. The imaging parameters will be TR/TE = 1200/82 ms, matrix size = 128 × 128, field of view (FOV) = 240 mm, slice thickness = 3 mm with no intersection gap, number of excitations = 2, and number of slices = 67.

The resting-state fMRI of 150 T2-weighted gradient echo planar imaging (EPI) will be acquired with 32 slices, with a resolution of 3 × 3 × 4 mm, and subjects will see a fixation cross (‘+’) passively at the centre of the screen. Image processing and analysis will be performed using FSL software packages (http://fsl.fmrib.ox.ac.uk/fsl/fslwiki/, accessed on 12 June 2022), and total brain an total grey matter will be extracted from the T1 structural scan. Voxel-based morphometry (VBM) will be used to segment and compare grey matter tissue maps and regional tissue density differences using [25]. Fractional anisotropy (FA) maps extracted from DTI imaging will be used to assess structural connectivity.

For evaluating functional connectivity, all resting state-fMRI (rs-fMRI) volumes will be pre-and post-processed, with motion correction and slice timing correction, and linearly registered to the Montreal Neurological Institute (MNI) standard space. The Rs-fMRI data analysis will use a data-driven approach. Multivariate Exploratory Linear Decomposition in FSL will be used to conduct an independent component analysis. A set of independent components will be identified as common resting-state functional networks. The global and local efficiency, modularity, and hubs will be computed using the Brain Connectivity Toolbox (https://sites.google.com/site/bctnet/, accessed on 12 June 2022). Between-group differences in the individual functional networks will be analysed by a dual regression approach. The significance threshold of the voxel-wise differences will be set at *p* < 0.05.

## 3. Statistical Analyses

Chi-squared tests and *t*-tests will be performed for the categorical and continuous variables, respectively, to examine the presence of any group differences in the demographics and clinical profiles between the treatment and sham groups at baseline. A two-way analysis of variance (ANOVA) with repeated measures will be used to examine the effect of time and treatment conditions on the CARS scores and other secondary outcomes across various assessment time points. Post hoc analyses will also be performed using Bonferroni corrected *t*-tests to investigate the improvements in all outcome variables in pairwise comparisons across various assessment time points. The level of statistical significance will be set at a *p* value < 0.05. Intention-to-treat last-observation-carried-forward scores will be used for the analyses. Missing data will be managed by multiple imputation. All computations will be performed using SPSS for Windows, version 25.0.

## 4. Conclusions

This study will be the first clinical trial in Hong Kong and nationwide to evaluate the efficacy and safety of TPS on adolescents with ASD. This study will contribute to the development of a new approach to facilitate ASD children in Hong Kong.

## Figures and Tables

**Figure 1 ijerph-19-15614-f001:**
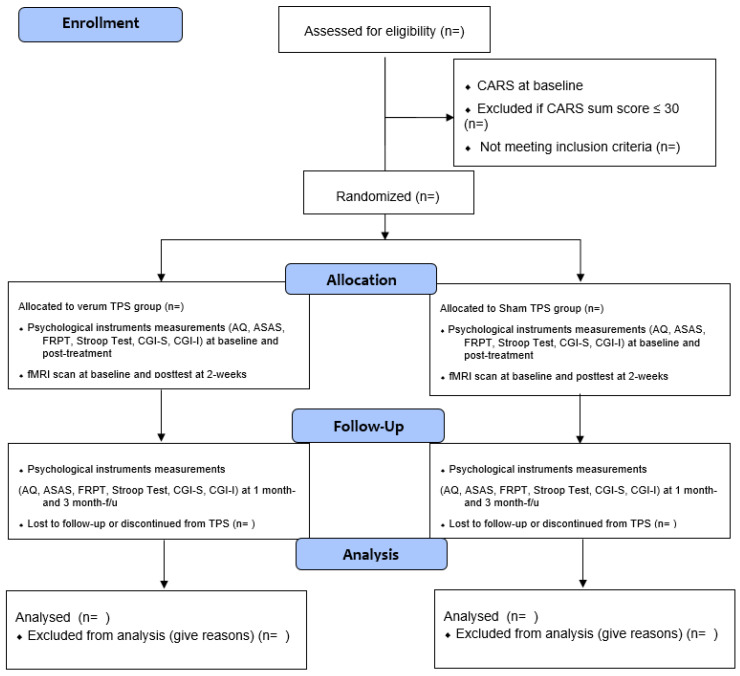
CONSORT flow diagram.

## Data Availability

The original contributions presented in the study are included in the article; further inquiries can be directed to the corresponding author.

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
