# Peer review of "Evaluating the Safety and Efficacy of Transcranial Pulse Stimulation on Autism Spectrum Disorder: A Double-Blinded, Randomized, Sham-Controlled Trial Protocol"

_ijerph, 2022, doi:10.3390/ijerph192315614_

Round 1
Reviewer 1 Report
The paper by Cheung et al. proposes a study protocol testing the safety and efficacy of Transcranial Pulse Stimulation (TPS) on young adolescents with Autism Spectrum Dis-3 order (ASD), by means of a two-armed (verum TPS group Vs sham TPS group), double-blinded, randomized, sham-controlled trial. The protocol proposed is well-thought, innovative and may greatly contribute to the treatment of ASD symptoms.
I have no major comments. The only issue that surprised me was not finding a reference to stimulation contraindications between the inclusion and exclusion criteria. Generally, when using NIBS, especially with clinical populations, it is always important to check that patients do not present any contraindications to stimulation. Are there any contraindications to TPS? Which ones are they? Will there be a screening to verify that all patients are eligible for stimulation?
I suggest revising the English of the entire manuscript. There are several typos (e.g. the abstract mentions 30 participants, but the power analysis predicts 36).
Author Response
Authors’ Response:
1: A more comprehensive description regarding the ‘exclusion criteria’ / contraindications on TPS stimulation has been added back on the main text (L133-136), highlighted in yellow fonts.
2: Yes, all subjects ought to pass through a screening tool ‘CARS’, fill in by subjects’ parents. All eligible subjects and their parents will then be zoom interviewed by a licensed UK and HK mental health practitioner (PI: Dr. Teris Cheung) in the recruitment phase for second screening to ensure that all subjects’ head/brain have no past history of trauma/tumour or any other medical complexities. Subjects also need to undertake pre-MRI (T1, DTI and resting-state MRI) to ensure that subjects’ brain have no signs of defects or abnormality as indicated in the exclusion criteria, and suitable for brain stimulation.
3: Sample size should be 34 instead. Main text had been amended (L140-145), highlighted in yellow fonts.
4: All typos have been rectified and highlighted in yellow fonts. English check was completed for the entire manuscript.
Thank you very much for your valuable comments.
Reviewer 2 Report
Dear authors,
the study protocol that you introduce in this manuscript is well desing, structure and written from a friendly reading point of view.
There some minor concerns related to typo issues and one major issue in relation to the sample size calculation.
Please, see file attached.
